# Sampling from Log-Concave Distributions with Infinity-Distance Guarantees

**Oren Mangoubi**
Worcester Polytechnic Institute

**Nisheeth K. Vishnoi**
Yale University

## Abstract

For a $d$-dimensional log-concave distribution $\pi(\theta) \propto e^{-f(\theta)}$ constrained to a convex body $K$, the problem of outputting samples from a distribution $\nu$ which is $\varepsilon$-close in infinity-distance $\sup_{\theta \in K} |\log \frac{\nu(\theta)}{\pi(\theta)}|$ to $\pi$ arises in differentially private optimization. While sampling within total-variation distance $\varepsilon$ of $\pi$ can be done by algorithms whose runtime depends polylogarithmically on $\frac{1}{\varepsilon}$, prior algorithms for sampling in $\varepsilon$ infinity distance have runtime bounds that depend polynomially on $\frac{1}{\varepsilon}$. We bridge this gap by presenting an algorithm that outputs a point $\varepsilon$-close to $\pi$ in infinity distance that requires at most $\mathrm{poly}(\log \frac{1}{\varepsilon}, d)$ calls to a membership oracle for $K$ and evaluation oracle for $f$, when $f$ is Lipschitz. Our approach departs from prior works that construct Markov chains on a $\frac{1}{\varepsilon^2}$-discretization of $K$ to achieve a sample with $\varepsilon$ infinity-distance error, and present a method to directly convert continuous samples from $K$ with total-variation bounds to samples with infinity bounds. This approach also allows us to obtain an improvement on the dimension $d$ in the running time for the problem of sampling from a log-concave distribution on polytopes $K$ with infinity distance $\varepsilon$, by plugging in TV-distance running time bounds for the Dikin Walk Markov chain.

## 1 Introduction

The problem of sampling from a log-concave distribution is as follows: For a convex body $K \subseteq \mathbb{R}^d$ and a convex function $f : K \to \mathbb{R}$, output a sample $\theta$ from the distribution $\pi(\theta) \propto e^{-f(\theta)}$. This is a basic problem in computer science, statistics, and machine learning, with applications to optimization and integration [1, 29], Bayesian statistics [39], reinforcement learning [6], and differential privacy [32, 20, 2, 24]. Sampling exactly from $\pi$ is known to be computationally hard for most interesting cases of $K$ and $f$ [16] and, hence, the goal is to output samples from a distribution $\nu$ that is at a small (specified) "distance" to $\pi$. For applications such as computing the integral of $\pi$, bounds in the total variation (TV) distance [1] or KL divergence (which implies a TV bound) are sufficient. In applications such as computing the expectation of a Lipschitz function with respect to $\pi$, Wasserstein distance may also be sufficient. In differentially private optimization [32, 20, 2, 19, 24], one requires bounds on the stronger infinity-distance –

$$\mathrm{d}_\infty(\nu, \pi) := \sup_{\theta \in K} \left| \log \frac{\nu(\theta)}{\pi(\theta)} \right|$$

– to guarantee pure differential privacy, and TV, KL, or Wasserstein bounds are insufficient; see [14].

Pure differential privacy (DP) is the strongest notion of DP and has been extensively studied (see e.g. the survey [14]). It has advantages over weaker notions of differential privacy. E.g., when privacy of "groups" of individuals (rather than just single individuals) must be preserved, any mechanism which is (pure) $\varepsilon$-DP (with respect to single individuals), is also $k\varepsilon$-DP with respect to subsets of $k$ individuals. Motivated by applications to differential privacy, we study the problem of designing efficient algorithms to output samples from a distribution $\nu$ which is $\varepsilon$-close in $\mathrm{d}_\infty$ to $\pi$.

36th Conference on Neural Information Processing Systems (NeurIPS 2022).

**Related works.** Several lines of work have designed Markov chains that generate samples from distributions that are close to a given log-concave distribution. These results differ in both their assumptions on the log-density and its support, as well as the distance used to measure closeness. One line of work includes bounds for sampling from a log-concave distribution on a compactly supported convex body within TV distance $O(\delta)$, including results with running time that is polylogarithmic in $\frac{1}{\delta}$ [1, 30, 29, 34] (as well as other results which give a running time bound that is polynomial in $\frac{1}{\delta}$ [18, 17, 5, 4]). In addition to assuming access to a value oracle for $f$, some Markov chains just need access to a membership oracle for $K$ [1, 30, 29], while others assume that $K$ is a given polytope: $\{\theta \in \mathbb{R}^d : A\theta \leq b\}$ [23, 33, 35, 34, 26]. They often also assume that $K$ is contained in a ball of radius $R$ and contains a ball of smaller radius $r$. Many of these results assume that the target log-concave distribution satisfies a "well-rounded" condition which says that the variance of the target distribution is $\Theta(d)$ [30, 29], or that it is in isotropic position (all eigenvalues of its covariance matrix are $\Theta(1)$) [25]; when applied to log-concave distributions that are not well-rounded or isotropic, these results require a "rounding" pre-processing procedure to find a linear transformation which makes the target distribution well-rounded or isotropic. Finally, it is often assumed that the function $f$ is such that $f$ is $L$-Lipschitz or $\beta$-smooth [34], including works handling the widely-studied special case when $f$ is uniform on $K$ where $L = \beta = 0$ (see e.g. [28, 23, 33, 35, 26, 8, 21]).

Another line of work gives sampling algorithms with bounds on the distance to the target density $\pi$ in terms of Wasserstein distance [13, 11], KL divergence [40, 12], and Renyi divergence [36]. In contrast to works which assume access to an oracle for the value of $f$, many of these results instead assume access to an oracle for the gradient of $f$ and require the log-density to be $L$-Lipschitz or $\beta$-smooth on all of $\mathbb{R}^d$ (or on, e.g., a cube containing $K$) for some $L, \beta > 0$. However, as noted earlier, bounds in the Wasserstein distance, KL divergence, and $\alpha$-Renyi divergence (for $\alpha < \infty$) also do not imply bounds on the infinity distance, and the running time bounds provided by these works are polynomial in $\frac{1}{\varepsilon}$. (See also Appendix A for additional discussion and challenges.)

Among prior works that give algorithms with bounds on $d_\infty$, [20] applies the grid walk Markov chain of [1] to sample from a uniform distribution on a convex body. [2] extends the approach of [20] to log-Lipschitz log-concave distributions. Unlike the TV-distance case where algorithms whose running time depends *logarithmically* on the error are known (e.g., [1, 29, 34]), the best available bounds for sampling within $O(\varepsilon)$ infinity-distance [20, 2] have runtime that is polynomial in $\frac{1}{\varepsilon}$ and a relatively large polynomial in $d$.

**Our contributions.** We present a new approach to output samples, which come with $d_\infty$ bounds, from a log-concave and log-Lipschitz distribution constrained to a a convex body. Specifically, when $K := \{\theta : A\theta \leq b\}$ is a polytope (where one is given $A$ and $b$) our main result (Theorem 2.1) guarantees samples from a distribution that is within $O(\varepsilon)$ error in $d_\infty$ and whose runtime depends logarithmically on $\frac{1}{\varepsilon}$ compared to the polynomial dependence of [2]. Our approach departs from prior works that construct Markov chains on a $\frac{1}{\varepsilon^2}$-discretization of $K$ to achieve a sample with $\varepsilon$ infinity-distance error, and we present a method (Algorithm 1) to directly convert continuous samples from $K$ with total-variation bounds to samples with infinity bounds (Theorem 2.2). This continuous-space approach also allows us to obtain an improvement on the dimension $d$ in the running time when $K$ is a polytope by plugging in TV-distance running time bounds for the Dikin Walk Markov chain of [34]. As immediate applications, we obtain faster algorithms for differentially private empirical risk minimization (Corollary 2.4) and low rank approximation (Corollary 2.5).

## 2 Results

Let $B(v, s) := \{z \in \mathbb{R}^d : \|z - v\|_2 \leq s\}$ and $\omega$ denote the matrix-multiplication constant.

**Theorem 2.1 (Main result)** *There exists an algorithm which, given $\varepsilon, L, r, R > 0$, $A \in \mathbb{R}^{m \times d}$, $b \in \mathbb{R}^m$ that define a polytope $K := \{\theta \in \mathbb{R}^d : A\theta \leq b\}$ contained in a ball of radius $R$, a point $a \in \mathbb{R}^d$ such that $K$ contains a ball $B(a, r)$ of smaller radius $r$, and an oracle for the value of a convex function $f : K \to \mathbb{R}^d$, where $f$ is $L$-Lipschitz, and defining $\pi$ to be the distribution $\pi \propto e^{-f}$, outputs a point from a distribution $\nu$ such that $d_\infty(\nu, \pi) < \varepsilon$. Moreover, with very high probability[1], this algorithm takes $O(T)$ function evaluations and $O(T \times md^{\omega-1})$ arithmetic operations, where $T = O((m^2d^3 + m^2dL^2R^2) \times [LR + d\log(\frac{Rd + LRd}{r\varepsilon})])$.*

---

[1]The number of steps is $O(\tau \times T)$, where $\mathbb{E}[\tau] \leq 3$, $\mathbb{P}(\tau \geq t) \leq \left(\frac{2}{3}\right)^t$ for $t \geq 0$, and $\tau \leq O(d\log(\frac{R}{r}) + LR)$ w.p. 1.

In comparison to the polynomial in $\frac{1}{\varepsilon}$ runtime bounds of [2], Theorem 2.1 guarantees a runtime that is *logarithmic* in $\frac{1}{\varepsilon}$, and also improves the dependence on the dimension $d$, in the setting where $K$ is a polytope. Specifically, [2] show that the number of steps of the grid walk to sample from $\pi$ with infinity-distance error at most $\varepsilon$ is

$$O\left(\frac{1}{\varepsilon^2}(d^{10} + d^6 L^4 R^4) \times \text{polylog}\left(\frac{1}{\varepsilon}, \frac{1}{r}, R, L, d\right)\right)$$

(Lemma 6.5 in the Arxiv version of [2]). When applying their algorithm to the setting where $f$ is constrained to a polytope $K = \{x \in \mathbb{R}^d : Ax \leq b\}$, each step of their grid walk Markov chain requires computing a membership oracle for $K$ and the value of the function $f$. The membership oracle can be computed in $O(md)$ arithmetic operations. Thus the bound on the number of arithmetic operations for each step of their grid walk is $O(md)$ (provided that each function evaluation takes at most $O(md)$ arithmetic operations). Thus the bound on the number of arithmetic operations to obtain a sample from $\pi$ is $O(\frac{1}{\varepsilon^2}(md^{11} + md^7 L^4 R^4) \times \text{polylog}(\frac{1}{\varepsilon}, \frac{1}{r}, R, L, d))$. Thus, Theorem 2.1 improves on this bound by a factor of roughly $\frac{1}{\varepsilon^2 m^3} d^{8-\omega}$. For example, when $m = O(d)$, as may be the case in differentially private applications, the improvement is $\frac{1}{\varepsilon^2} d^{5-\omega}$.

We note that the bounds of [2] also apply in the more general setting where $K$ is a convex body with membership oracle. One can extend our bounds to achieve a runtime that is logarithmic in $\frac{1}{\varepsilon}$ (and polynomial in $d, L, R$) in the more general setting where $K$ is a convex body with membership oracle; we omit the details (see Remark 2.3).

Moreover, we also note that while there are several results which achieve $O(\delta)$ TV bounds in time logarithmic in $\frac{1}{\delta}$, TV bounds do not in general imply $O(\varepsilon)$ bounds on the KL or Renyi divergence, or on the infinity-distance, for any $\delta > 0$.[2] On the other hand, an $\varepsilon$-infinity-distance bound does immediately imply a bound of $\varepsilon$ on the KL divergence $D_{\text{KL}}$, and $\alpha$-Renyi divergence $D_\alpha$, since $D_{\text{KL}}(\mu, \pi) \leq d_\infty(\mu, \pi)$ and $D_\alpha(\mu, \pi) \leq d_\infty(\mu, \pi)$ for any $\alpha > 0$ and any pair of distributions $\mu, \pi$. Thus, under the same assumptions on $K$ and $f$, Theorem 2.1 implies a method of sampling from a Lipschitz concave log-density on $K$ with $\varepsilon$ KL and Renyi divergence error in a number of arithmetic operations that is *logarithmic* in $\frac{1}{\varepsilon}$, with the same bound on the number of arithmetic operations.

The polynomial dependence on $\frac{1}{\varepsilon}$ in [2] is due to the fact that they rely on a discrete-space Markov chain [1], on a grid with cells of width $w = O(\frac{\varepsilon}{L\sqrt{d}})$, to sample from $\pi$ within $O(\varepsilon)$ infinity-distance. Since their Markov chain's runtime bound is polynomial in $w^{-1}$, they get a runtime bound for sampling within $O(\varepsilon)$ infinity-distance that is polynomial in $\frac{1}{\varepsilon}$. The proof of Theorem 2.1 bypasses the use of discrete grid-based Markov chains by introducing Algorithm 1 which transforms any sample within $\delta = O(\varepsilon e^{-d-nLR})$ TV distance of the distribution $\pi \propto e^{-f}$ on the *continuous* set $K$ (as opposed to a discretization of $K$), into a sample within $O(\varepsilon)$ infinity-distance from $\pi$. This allows us to make use of a continuous-space Markov chain, whose step size is not restricted to a grid of width $O(\frac{\varepsilon}{L\sqrt{d}})$ and is instead independent of $\varepsilon$, to obtain a sample within $O(\varepsilon)$ infinity-distance from $\pi$ in time that is logarithmic in $\frac{1}{\varepsilon}$.

**Theorem 2.2 (Main technical contribution: Converting TV bounds to infinity-distance bounds)**
*There exists an algorithm (Algorithm 1) which, given $\varepsilon, r, R, L > 0$, a membership oracle for a convex body $K$ contained in a ball of radius $R$ and containing a ball $B(0, r)$, and an oracle which outputs a point from a distribution $\mu$ which has TV distance*

$$\delta \leq O\left(\varepsilon \times \left(\frac{R(d\log(R/r) + LR)^2}{\varepsilon r}\right)^{-d} e^{-LR}\right)$$

*from a distribution $\pi \propto e^{-f}$ where $f : K \to \mathbb{R}$ is an L-Lipschitz function (see Appendix B for the exact values of $\delta$ and related hyper-parameters), outputs a point $\hat{\theta} \in K$ such that the distribution $\nu$*

---

[2] For instance, if $\pi(\theta) = 1$ with support on $[0, 1]$, for every $\delta > 0$ there is a distribution $\nu$ where $\|\nu - \pi\|_{\text{TV}} \leq 2\delta$ and yet $d_\infty(\nu, \pi) \geq D_{\text{KL}}(\nu, \pi) \geq \frac{1}{2}$. ($\nu(\theta) = e^{\frac{1}{\delta}}$ on $\theta \in [0, \delta e^{-\frac{1}{\delta}}]$, $\nu(\theta) = \frac{1-\delta}{1-\delta e^{-\frac{1}{\delta}}}$ on $(\delta e^{-\frac{1}{\delta}}, 1]$ and $\nu(\theta) = 0$ otherwise)

*of $\hat{\theta}$ satisfies* $\mathrm{d}_\infty(\nu, \pi) \leq \varepsilon$. *Moreover, with very high probability*[3], *this algorithm finishes in* $O(1)$ *calls to the sampling and membership oracles, plus* $O(d)$ *arithmetic operations.*

To the best of our knowledge Theorem 2.2 is the first result which for any $\varepsilon, L, r, R > 0$, when provided as input a sample from a continuous-space distribution on a convex body $K$ within some TV distance $\delta = \delta(\varepsilon, r, R, L) > 0$ from a given $L$-log-Lipschitz distribution $\pi$ on $K$, where $K$ is contained in a ball of radius $R$ and containing a ball of smaller radius $r$, outputs a sample with distribution within infinity-distance $O(\varepsilon)$ from $\pi$. This is in contrast to previous works [2] (see also [20] which applies only to the special case where $\pi$ is the uniform distribution on $K$) which instead require as input a sample with bounded TV distance from the restriction of $\pi$ on a *discrete* grid on $K$, and then convert this discrete-space sample into a sample within infinity-distance $O(\varepsilon)$ from the continuous-space distribution $\pi : K \to \mathbb{R}$.

---

**Algorithm 1:** Interior point TV to infinity-distance converter

**Input:** $d \in \mathbb{N}$
**Input:** A membership oracle for a convex body $K \in \mathbb{R}^d$ and an $r > 0$ such that $B(0, r) \subseteq K$.
**Input:** A sampling oracle which outputs a point from a distribution $\mu : K \to \mathbb{R}$
**Output:** A point $\hat{\theta} \in K$.
1 **Hyperparameters:** $\Delta > 0$, $\tau_{\max} \in \mathbb{N}$ (set in Appendix B)
2 **for** $i = 1, \ldots, \tau_{\max}$ **do**
3      Sample a point $\theta \sim \mu$
4      Sample a point $\xi \sim \mathrm{Unif}(B(0, 1))$
5      Set $Z \leftarrow \theta + \Delta r \xi$
6      Set $\hat{\theta} \leftarrow \frac{1}{1-\Delta} Z$
7      If $\hat{\theta} \in K$, output $\hat{\theta}$ with probability $\frac{1}{2}$ and halt. Otherwise, continue.
8 **end**
9 Sample a point $\hat{\theta} \sim \mathrm{Unif}(B(0, r))$
10 Output $\hat{\theta}$

---

**Remark 2.3 (Extension to convex bodies with membership oracles)** *We note that Theorem 2.1 can be extended to the general setting where $K$ is an arbitrary convex body in a ball of radius $R$ and containing a ball of smaller radius $r$, and we only have membership oracle access to $K$. Namely, one can plug in the results of [29] to our Theorem 2.2 to generate a sample from a $L$-Lipschitz concave log-density on an arbitrary convex body $K$ in a number of operations that is (poly)-logarithmic in $\frac{1}{\varepsilon}, \frac{1}{r}$ and polynomial on $d, L, R$. We omit the details.*

**Applications to differentially private optimization.** Sampling from distributions with $O(\varepsilon)$ infinity-distance error has many applications to differential privacy. Here, the goal is to find a randomized mechanism $h : \mathcal{D}^n \to \mathcal{R}$ which, given a dataset $x \in \mathcal{D}^n$ consisting of $n$ datapoints, outputs model parameters $\hat{\theta} \in \mathcal{R}$ in some parameter space $\mathcal{R}$, which minimize a given (negative) utility function $f(\theta, x)$, under the constraint that the output $\hat{\theta}$ preserves the pure $\varepsilon$-differential privacy of the data points $x$. A randomized mechanism $h : \mathcal{D}^n \to \mathcal{R}$ is said to be $\varepsilon$-differentially private if for any datasets $x, x' \in \mathcal{D}$ which differ by a single datapoint, and any $S \subseteq \mathcal{R}$, we have that

$$\mathbb{P}(h(x) \in S) \leq e^\varepsilon \mathbb{P}(h(x') \in S);$$

see [14].

As one application of Theorem 2.1, we consider the problem of finding an (approximate) minimum $\hat{\theta}$ of an empirical risk function $f : K \times \mathcal{D}^n \to \mathbb{R}$ under the constraint that the output $\hat{\theta}$ is $\varepsilon$-differentially private, where $f(\theta, x) := \sum_{i=1}^n \ell_i(\theta, x_i)$. Following [2], we assume that the $\ell_i(\cdot, x)$ are $L$-Lipschitz for all $x \in \mathcal{D}^n$, $i \in \mathbb{N}$, for some given $L > 0$. In this setting [2] show that the minimum ERM utility bound under the constraint that $\hat{\theta}$ is pure $\varepsilon$-differentially private, $\mathbb{E}_{\hat{\theta}}[f(\hat{\theta}, x)] - \min_{\theta \in K} f(\theta, x) = \Theta(\frac{dLR}{\varepsilon})$, is achieved if one samples $\hat{\theta}$ from the exponential mechanism $\pi \propto e^{-\frac{\varepsilon}{2LR} f}$ with infinity-distance error at most $O(\varepsilon)$. Plugging Theorem 2.1 into the framework of

---

[3]Algorithm 1 finishes in $\tau$ calls to the sampling and membership oracles, plus $O(\tau d)$ arithmetic operations, where $\mathbb{E}[\tau] \leq 3$ and $\mathbb{P}(\tau \geq t) \leq \left(\frac{2}{3}\right)^t$ for all $t \geq 0$ and $\tau \leq 5d \log(\frac{R}{r}) + 5LR + 2$ w.p. 1.

the exponential mechanism, we obtain a pure $\varepsilon$-differentially private mechanism which achieves the minimum expected risk (Corollary 2.4, see Section C.1 for a proof).

**Corollary 2.4 (Differentially private empirical risk minimization)** *There exists an algorithm which, given $\varepsilon, L, r, R > 0$, $A \in \mathbb{R}^{m \times d}$, $b \in \mathbb{R}^m$ that define a polytope $K := \{\theta \in \mathbb{R}^d : A\theta \leq b\}$ contained in a ball of radius $R$ and containing a ball $B(0, r)$ of smaller radius $r$, and a convex function $f(\theta, x) := \sum_{i=1}^n \ell_i(\theta, x_i)$, where each $\ell_i : K \to \mathbb{R}$ is $L$-Lipschitz, outputs a random point $\hat{\theta} \in K$ which is pure $\varepsilon$-differentially private and satisfies*

$$\mathbb{E}_{\hat{\theta}}[f(\hat{\theta}, x)] - \min_{\theta \in K} f(\theta, x) \leq O\left(\frac{dLR}{\varepsilon}\right).$$

*Moreover, this algorithm takes at most $T \times md^{\omega-1}$ arithmetic operations plus $T$ evaluations of the function $f$, where $T = O\left((m^2 d^3 + m^2 dn^2 \varepsilon^2) \times (\varepsilon n + d)\log^2(\frac{nRd}{r\varepsilon})\right)$.*

Corollary 2.4 improves on the previous bound [2] of $O((\frac{1}{\varepsilon^2}(m + n)d^{11} + \varepsilon^2 n^4(m + n)d^7) \times \text{polylog}(\frac{nRd}{r\varepsilon})))$ arithmetic operations by a factor of roughly $\max\left(\frac{d^{8-\omega}}{\varepsilon^2 m^2}, \frac{1}{\varepsilon m^2}nd^5\right)$, in the setting where the $\ell_i$ are $L$-Lipschitz on a polytope $K$ and each $\ell_i$ can be evaluated in $O(d)$ operations. See Appendix C.1 for a proof of this corollary.

As another application, we consider the problem of finding a low-rank approximation of a sample covariance matrix $\Sigma = \sum_{i=1}^n u_i u_i^\top$ where the datapoints $u_i \in \mathbb{R}^d$ satisfy $\|u_i\| \leq 1$, in a differentially private manner. Given any $k > 0$, the goal is to find a (random) rank-$k$ projection matrix $P$ which maximizes the average variance $\mathbb{E}_P[\langle \Sigma, P \rangle]$ of the matrix $\Sigma$ (also reffered to as the utility of $P$), under the constraint that the mechanism which outputs the matrix $P$ is $\varepsilon$-differentially private. This problem has many applications to statistics and machine learning, including differentially private principal component analysis (PCA) [7, 3, 15, 24].

When privacy is not a concern, the solution $P$ which maximizes the variance is just a projection matrix onto the subspace spanned by top-$k$ eigenvectors of $\Sigma$, and the maximum variance satisfies $\langle \Sigma, P \rangle = \sum_{i=1}^k \lambda_i$, where $\lambda_1 \geq \cdots \geq \lambda_d > 0$ denote the eigenvalues of $\Sigma$. However, when privacy is a concern, there is a tradeoff between the desired privacy level $\varepsilon$ and the utility $\mathbb{E}_P[\langle \Sigma, P \rangle]$, and the maximum utility $\mathbb{E}_P[\langle \Sigma, P \rangle]$ one can achieve decreases with the privacy parameter $\varepsilon$. The best current utility bound for an $\varepsilon$-differentially private low rank approximation algorithm was achieved in [24], who show that one can find a pure $\varepsilon$-differentially private random rank-$k$ projection matrix $P$ such that $\mathbb{E}_P[\langle \Sigma, P \rangle] \geq (1 - \delta) \sum_{i=1}^k \lambda_i$ whenever $\sum_{i=1}^k \lambda_i \geq \Omega\left(\frac{dk}{\varepsilon\delta} \log \frac{1}{\delta}\right)$ for any $\delta > 0$. To generate the matrix $P$, their algorithm generates a sample, with infinity-distance error $O(\varepsilon)$, from a Lipschitz concave log-density on a polytope, and transforms this sample into a projection matrix $P$. The sampling algorithm used in [24] has a bound of $\text{poly}(\frac{1}{\varepsilon}, d, \lambda_1 - \lambda_d)$ arithmetic operations and they leave as an open problem whether this can be improved from a polynomial dependence on $\frac{1}{\varepsilon}$ to a logarithmic dependence on $\frac{1}{\varepsilon}$. Corollary 2.5 shows that a direct application of Theorem 2.1 resolves this problem. (See Section C.2 for a proof.)

**Corollary 2.5 (Differentially private low rank approximation)** *There exists an algorithm which, given a sample covariance matrix $\Sigma = \sum_{i=1}^n u_i u_i^\top$ for datapoints $u_i \in \mathbb{R}^d$ satisfying $\|u_i\| \leq 1$, its eigenvalues $\lambda_1 \geq \ldots \lambda_d > 0$, an integer $k$, and $\varepsilon, \delta > 0$, outputs a random rank-k symmetric projection matrix $P$ such that $P$ is $\varepsilon$-differentially private and satisfies the utility bound*

$$\mathbb{E}_P[\langle \Sigma, P \rangle] \geq (1 - \delta) \sum_{i=1}^k \lambda_i(\Sigma)$$

*whenever $\sum_{i=1}^k \lambda_i(\Sigma) \geq C\frac{dk}{\varepsilon\delta} \log \frac{1}{\delta}$ for some universal constant $C > 0$. Moreover the number of arithmetic operations is logarithmic in $\frac{1}{\varepsilon}$ and polynomial in $d$ and $\lambda_1 - \lambda_d$.*

## 3 Proof Overviews

Given any $\varepsilon$, and a function $f : \mathbb{R}^d \to \mathbb{R}$, the goal is to sample from a distribution $\pi(\theta) \propto e^{-f(\theta)}$, constrained to a $d$-dimensional convex body $K$ with infinity-distance error at most $O(\varepsilon)$ in a number

of arithmetic operations that is logarithmic in $\frac{1}{\varepsilon}$. We assume that $K$ is contained in a ball of some radius $R > 0$ and contains a ball of some radius $r > 0$, and $f$ is $L$-Lipschitz. In addition we would also like our bounds to be polylogarithmic in $\frac{1}{r}$, and polynomial in $d, L, R$ with a lower-order dependence on the dimension $d$ than currently available bounds for sampling from Lipschitz concave log-densities on a polytope in infinity-distance [2]. We note that since whenever $K$ is contained in a ball of radius $R$ and contains a ball $B(0, r)$ of smaller radius $r$, we also have that $B(0, r) \subseteq K \subseteq B(0, 2R)$, without loss of generality, we may assume that $B(0, r) \subseteq K \subseteq B(0, R)$ as this would only change the bounds provided in our main theorems by a constant factor.

The main ingredient in the proof of Theorem 2.1 is Theorem 2.2 that uses Algorithm 1 to transform a TV-bounded sample into a sample from $\pi$ with error bounded in $\mathrm{d}_\infty$. Subsequently, we invoke Theorem 2.1 when $K$ is given as a polytope $K := \{x \in \mathbb{R}^d : Ax \leq b\}$ and plug in the Dikin Walk Markov chain of [34] which generates independent samples from $\pi$ with bounded TV error. We first present an overview of the proof of Theorem 2.2. (The full proof has been omitted to space restrictions and presented in Appendix B.) The proof of Theorem 2.1 is presented in Section 3.2.

### 3.1 Converting samples with TV bounds to infinity-distance bounds; proof of Theorem 2.2

**Impossibility of obtaining log-dependence on infinity-distance via grid walk.** One approach is to observe that if $e^{-f}$ has support on a discrete space $S$ with at most $|S|$ points, then any $\nu$ such that $\|\nu - \pi\|_{\mathrm{TV}} \leq \varepsilon$ also satisfies

$$\mathrm{d}_\infty(\nu, \pi) \leq 2|S|\frac{\max_{z \in S} e^{-f(z)}}{\min_{z \in S} e^{-f(z)}} \times \varepsilon$$

for any $\varepsilon \leq \min_{z \in S} \pi(z)$. This suggests forming a grid $G$ over $K$, then using a discrete Markov chain to generate a sample $\theta$ within $O(\varepsilon)$ TV distance of the discrete distribution $\pi_G \propto e^{-f}$ with support on the grid $G$, and then designing an algorithm which takes as input $\theta$ and outputs a point with bounded infinity-distance to the continuous distribution $\pi$. This approach was used in [20] in the special case when $\pi$ is uniform on $K$, and then extended by [2] to log-Lipschitz log-concave distributions. In their approach, [2] first run a "grid-walk" Markov chain on a discrete grid in a cube containing $K$. They then apply the bound from [1] which says that the grid walk obtains a sample $Z$ within TV distance $O(\delta)$ from the distribution $\propto e^{-f}$ (restricted to the grid) in time that is polylogarithmic in $\frac{1}{\delta}$ and quadratic in $a^{-1}$, where $a$ is the distance between neighboring grid points. Since their grid has size $|S| = \Theta((\frac{R}{a})^d)$, a TV distance of $O(\delta)$ automatically implies an infinity-distance of $O(\delta c|S|)$, where $c$ is the ratio of the maximum to the minimum probability mass satisfying $c \leq e^{LR}$ since $f$ is $L$-Lipschitz on $K \subseteq B(0, R)$. Thus, by using the grid walk to sample within TV-distance $\delta = O\left(\frac{\varepsilon}{|S|c}\right)$ from the discrete distribution $\pi_G$, they obtain a sample $Z$ which also has infinity-distance $O(\varepsilon)$ from $\pi_G$. Finally, to obtain a sample from the distribution $\pi \propto e^{-f}$ on the continuous space $K$, they sample a point uniformly from the "grid cell" $[Z - a, Z + a]^d$ centered at $Z$. Since $f$ is $L$-Lipschitz, the ratio $\frac{e^{-f(Z)}}{e^{-f(w)}}$ is bounded by $O(\varepsilon)$ for all $w$ in the grid cell $[Z - a, Z + a]^d$ as long as $a = O\left(\frac{\varepsilon}{L\sqrt{d}}\right)$, implying that the sample is an infinity-distance of $O(\varepsilon)$ from $\pi \propto e^{-f}$. However, since the running time bound of the grid walk is quadratic in $a^{-1}$, the grid coarseness $a = O\left(\frac{\varepsilon}{L\sqrt{d}}\right)$ needed to achieve $O(\varepsilon)$ infinity-distance from $\pi$ leads to a running time bound which is *quadratic* in $\frac{1}{\varepsilon}$.

To get around this problem, rather than relying on the use of a discrete-space Markov chain such as the grid walk to sample within $O(\varepsilon)$ infinity-distance from $\pi$, we introduce an algorithm (Algorithm 1) which transforms any sample within $\delta = O\left(\varepsilon e^{-d - LR}\right)$ TV distance from the distribution $\pi \propto e^{-f}$ on the *continuous* space $K$ (as opposed to a grid-based discretization of $K$), into a sample within $O(\varepsilon)$ infinity-distance from $\pi$. This allows us to make use of a continuous-space Markov chain, such as the Dikin walk of [34], whose step-size is not restricted by a grid of coarseness $w = O\left(\frac{\varepsilon}{L\sqrt{d}}\right)$ and instead is independent of $\varepsilon$, in order to generate a sample within $O(\varepsilon)$ infinity-distance from $\pi$ in runtime that is *logarithmic* in $\frac{1}{\varepsilon}$.

**Converting continuous space TV-bounded samples to infinity-distance bounded samples.** As discussed in Section 1, there are many Markov chain results which allow one to sample from a log-

concave distribution on $K$ with error bounded in weaker metrics such as total variation, Wasserstein, or KL divergence. However, when sampling from a continuous distribution, bounds in these metrics do not directly imply bounds in infinity-distance. And techniques used to prove bounds in weaker metrics do not easily extend to methods for bounding the infinity-distance; see Section A.1.

*Convolving with continuous noise.* As a first attempt, we consider the following simple algorithm: sample a point $\theta \sim \mu$ from a distribution $\mu$ with total variation error $\|\mu - \pi\|_{\mathrm{TV}} \leq O(\varepsilon)$. Since $f$ is $L$-Lipschitz, for any $\Delta < \frac{\varepsilon}{L}$ and any ball $B(z, \Delta)$ in the $\Delta$-interior of $K$ (denoted by $\mathrm{int}_\Delta(K)$; see Definition B.1), we can obtain a sample from a distribution $\nu$ such that $\log \left( \frac{\nu(z)}{\mu(z)} \right) \leq \varepsilon$ for all $z \in \mathrm{int}_\Delta(K)$ by convolving $\mu$ with the uniform distribution on the ball $B(0, \Delta)$. Sampling from this distribution $\nu$ can be achieved by first sampling $\theta \sim \mu$ and then adding noise $\xi \sim \mathrm{Unif}(B(0, \Delta))$ to the sample $\theta$.

Unfortunately, this simple algorithm does not allow us to guarantee that $\log(\frac{\nu(z)}{\mu(z)}) \leq \varepsilon$ at points $z \notin \mathrm{int}_\Delta(K)$ which are a distance less than $\Delta$ from the boundary of $K$. To see why, suppose that $K = [0,1]^d$ is the unit cube, that $f$ is constant on $K$, and consider a point $w = (1, \ldots, 1)$ at the corner of the cube $K$. In this case we could have that $\nu(z) \leq 2^{-d}\pi(z)$ for all $z$ in some ball containing $w$, and hence $\mathrm{d}_\infty(\nu, \pi) = \sup \left| \log(\frac{\nu(z)}{\pi(z)}) \right| \geq d \log(2)$, no matter how small we make $\Delta$.

*Stretching the convex body to handle points close to the boundary.* To get around this problem, we would like to design an algorithm which samples from some distribution $\nu$ such that $\left| \log \frac{\nu(z)}{\pi(z)} \right| \leq \varepsilon$ for all $z \in K$, including at points $z$ near the corners of $K$. Towards this end, we first consider the special case where $K$ is itself contained in the $\Delta$-interior of another convex body $K'$, the function $f : K \to \mathbb{R}$ extends to an $L$-Lipschitz function on $K'$ (also referred to here with slight abuse of notation as $f$) and where we are able to sample from the distribution $\propto e^{-f}$ on $K'$ with $O(\varepsilon)$ total variation error. If we sample $\theta \sim e^{-f}$ on $K'$ with total variation error $O(\delta)$ where $\delta \leq \varepsilon e^{-d \log(R)}$, add noise $\xi \sim \mathrm{Unif}(B(0, \Delta))$ to $\theta$ for $\Delta = \frac{\delta}{LR}$, and then reject $\theta + \xi$ only if it is not in $K$, we obtain a sample whose distribution is $O(\varepsilon)$ from the distribution $\propto e^{-f}$ on $K$ in infinity-distance.

However, we would still need to define and sample from such a convex body $K'$, and to make sure that $K'$ is not too large when compared to $K$; otherwise the samples from the distribution $\propto e^{-f}$ on $K'$ may be rejected with high probability. Moreover, another issue we need to deal with is that $f$ may not even be defined outside of $K$.

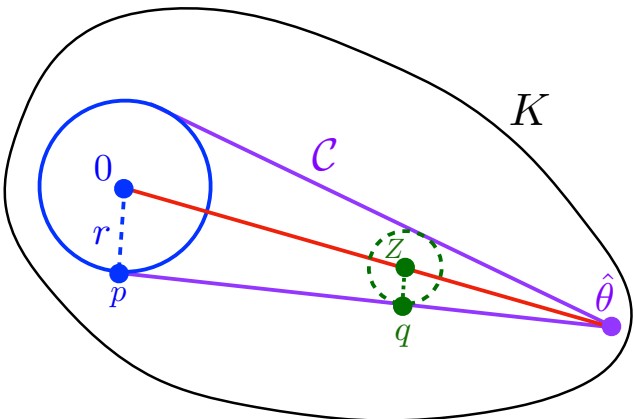

Figure 1: The construction used in the proof of Lemma B.1.

To get around these two problems, in Algorithm 1, we begin by taking as input a point $\theta \sim \mu$ sampled from some distribution $\mu$ supported on $K$ where $\|\mu - \pi\|_{\mathrm{TV}} \leq \delta$ for some $\delta \leq \varepsilon e^{-d \log(R)}$, and add noise $\xi \sim \mathrm{unif}(B(0, \Delta r))$ in order to sample from a distribution $\hat{\mu}$ which satisfies $\left| \log \frac{\hat{\mu}(z)}{\pi(z)} \right| \leq \varepsilon$ for all $z \in \mathrm{int}_{\Delta r}(K)$. Here $r$ is the radius of the small ball contained in $K$; the choice of radius $\Delta r$ for the noise distribution is because we will show in the following paragraphs that to sample from

the distribution $\pi$ on $K$ with infinity-distance error $\varepsilon$ it is sufficient sample a point in the $\Delta r$-interior of $K$ and to then apply a "stretching" operation to $K$.

We still need a method of sampling within $O(\varepsilon)$ infinity-distance error of $\pi$ on all of $K$, including in the region $K \backslash \mathrm{int}_{\Delta r}(K)$ near the boundary of $K$. Towards this end, after Algorithm 1 generates a point $Z = \theta + \xi$ from the above-mentioned distribution $\hat{\mu}$, it then multiplies $Z$ by $\frac{1}{1-\Delta}$ and returns the resulting point $\hat{\theta} := \frac{1}{1-\Delta}Z$ if it is $K$, in other words, if $Z \in (1-\Delta)K$. If we can show that $(1-\Delta)K \subseteq \mathrm{int}_{\Delta r}(K)$, then this would imply that $\left| \log \frac{\hat{\mu}(z)}{\pi(z)} \right| \leq \varepsilon$ for all $z \in (1-\Delta)K$, and hence that the distribution of $\hat{\nu}$ of the returned point $\hat{\theta}$ satisfies

$$\left| \log \frac{\hat{\nu}(z)}{\pi((1-\Delta)z)} \right| \leq \left| \log \frac{\hat{\mu}((1-\Delta)z)}{\pi((1-\Delta)z)} \right| + \log \frac{1}{(1-\Delta)^d} \leq O(\varepsilon)$$

for all $z \in K$. Since $f$ is $L$-Lipschitz we have $\left| \log \frac{\pi(\theta)}{\pi((1-\Delta)\theta)} \right| = O(\varepsilon)$ for all $\theta \in K$, and hence we would then have that the distribution $\hat{\nu}$ of the point $\hat{\theta}$ returned by Algorithm 1 satisfies

$$\left| \log \frac{\hat{\nu}(z)}{\pi(z)} \right| = \left| \log \frac{\hat{\nu}(z)}{\pi((1-\Delta)z)} \right| + O(\varepsilon) \leq O(\varepsilon) \qquad \forall z \in K. \tag{1}$$

However, for (1) to hold, we still need to show that $(1-\Delta)K \subseteq \mathrm{int}_{\Delta r}(K)$ (proved in Lemma B.1). In other words, we would like to show that for any point $Z \in (1-\Delta)K$, there is a ball $B(Z, \Delta r)$ centered at $Z$ of radius $\Delta r$ contained in $K$. To show this fact, it is sufficient to consider the convex hull $\mathcal{C}$ of $B(0, r) \cup \left\{ \frac{1}{1-\Delta}Z \right\} \subseteq K$, and show that it contains the ball $B(Z, \Delta r)$. Towards this end, we make the following geometric construction: we let $p$ be a point such that the line $p\hat{\theta}$ is tangent to $B(0, r)$, and $q$ the point on $p\hat{\theta}$ which minimizes the distance $\|q - Z\|_2$ (see Figure 1). Since $\angle 0p\hat{\theta}$ and $\angle Zq\hat{\theta}$ are both right angles, we have that the triangles $0p\hat{\theta}$ and $Zq\hat{\theta}$ are similar triangles, and hence that $\frac{\|Z-q\|_2}{r} = \frac{\|Z-\hat{\theta}\|_2}{\|\hat{\theta}-0\|_2}$. In other words,

$$\|Z - q\|_2 = \frac{\|Z - \hat{\theta}\|_2}{\|\hat{\theta} - 0\|_2} \times r = \frac{\left\| Z - \frac{1}{1-\Delta}Z \right\|_2}{\left\| \frac{1}{1-\Delta}Z \right\|_2} \times r = \Delta \times r,$$

implying a ball of radius $\Delta r$ centered at $Z$ is contained in $\mathcal{C} \subseteq K$, and hence that

$$(1-\Delta)K \subseteq \mathrm{int}_{\Delta r}(K).$$

*Bounding the infinity distance error.* To complete the bound on the infinity-distance of the distribution $\hat{\nu}$ of the point returned by Algorithm 1 to the target distribution $\pi$, we must show both a lower bound (Lemma B.5) and an upper bound (Lemma B.6) on the ratio $\frac{\hat{\nu}(\theta)}{\pi(\theta)}$ at every point $\theta \in K$. Both the upper and lower bounds are necessary to bound the infinity-distance $\mathrm{d}_\infty(\hat{\nu}(\theta), \pi(\theta)) = \sup_{\theta \in K} \left| \log \frac{\hat{\nu}(\theta)}{\pi(\theta)} \right|$.

Both Lemmas B.5 and B.6 require the input point to have TV error $\delta < \varepsilon(\frac{R}{\Delta r})^{-d} e^{-LR}$. The term $(\frac{R}{\Delta r})^{-d}$ is a lower bound on the ratio of the volume of $K$ to the volume of the smoothing ball $B(0, \Delta r)$; this bound holds since $K$ is contained in a ball of radius $R$. The term $e^{-LR}$ is a lower bound on the ratio $\frac{\min_{w \in K} \pi(w)}{\max_{w \in K} \pi(w)}$ of the minimum value of the density $\pi$ to the maximum value of $\pi$ at any two points in $K$; this bound holds since $f$ is $L$-Lipschitz.

The above choice of $\delta$ ensures that in any ball $B(z, \Delta r)$ with center $z$ in the $\Delta r$-interior of $K$, the distribution $\mu$ of the input point, which satisfies $\|\mu - \pi\|_{\mathrm{TV}} \leq \delta$, will have between $e^{-\varepsilon}$ and $e^\varepsilon$ times the probability mass which the target distribution $\pi$ has inside the ball $B(z, \Delta r)$. Thus, when the distribution $\mu$ is smoothed by adding noise uniformly distributed on a ball of radius $\Delta r$, the smoothed distribution $\tilde{\nu}(\theta)$ is within $e^{-\varepsilon}$ and $e^\varepsilon$ times the target probability density $\pi(\theta)$ at any point $\theta$ in the $\Delta r$-interior of $K$, allowing us to bound the infinity distance error of the smoothed distribution $\tilde{\nu}$ at any point $\theta$ in the $\Delta r$-interior of $K$. We then apply this fact, together with Lemma

B.1 which says that for any point $\theta \in K$ the point $(1-\Delta)\theta$ is in the $\Delta r$-interior of $K$, to bound the distribution $\nu$ of the output point (after the stretching operation) as follows,

$$\nu(\theta) \geq (1-\Delta)^d \tilde{\nu}((1-\Delta)\theta) \overset{\text{Lemma B.1}}{\geq} (1-\Delta)^d \pi((1-\Delta)\theta) \times e^{-\varepsilon} \geq \pi(\theta)e^{-\frac{\varepsilon}{2}}. \qquad (2)$$

Here our choice of hyperparameter $\Delta \leq \frac{\varepsilon}{\max(d, LR)}$ ensures that $(1-\Delta)^d = \Omega(1)$ and, since $f$ is $L$-Lipschitz, that $\pi((1-\Delta)\theta) \geq e^{-\varepsilon}\pi(\theta)$. This proves the lower bound (Lemma B.5). The proof of the upper bound (Lemma B.6) follows in a similar way as equation (2) but with the inequalities going in the opposite direction.

*Bounding the number of iterations and concluding the proof of Theorem 2.2.* We still need to deal with the problem that the point $\hat{\theta}$ may not be accepted. If this occurs, roughly speaking, we repeat the above procedure until a point $\hat{\theta}$ with distribution $\hat{\nu}$ is accepted. To bound the number of iterations, we show that $\hat{\theta}$ is in $K$ with high probability. Towards this end, we first use the facts that $f$ is $L$-Lipschitz and $K \subseteq B(0, R)$, to show that the probability a point sampled from $\pi \propto e^{-f}$ lies inside $(1-\Delta)K$ is at least $(1-\Delta)^d e^{-L\Delta R} \geq \frac{9}{10}$ (Lemma B.2). Lemma B.2 says that if you stretch the polytope by a factor of $\frac{1}{1-\Delta}$, then most of the volume of the stretched polytope $(\frac{1}{1-\Delta})K$ remains inside the original polytope $K$. The term $(1-\Delta)^d$ is just the ratio of the volume of $(1-\Delta)K$ to the volume of $K$. And, since $f$ is $L$-Lipschitz, the term $e^{-L\Delta R}$ bounds the ratio $\frac{\pi(\theta)}{\pi(\frac{1}{1-\Delta}\theta)}$ of the target density at any point $\theta \in K$ to the value of $\pi$ at the point $\frac{1}{1-\Delta}\theta$ to which the stretching operation transports $\theta$, whenever $\frac{1}{1-\Delta}\theta \in K$. The choice of hyperparameter $\Delta \leq \frac{\varepsilon}{\max(d, LR)}$ ensures that the acceptance probability $(1-\Delta)^d e^{-L\Delta R}$ guaranteed by Lemma B.2 is at least $\frac{9}{10}$. Since the convex body $(1-\Delta)K$ contains the ball $B(0, \frac{r}{2})$, applying Lemma B.1 a second time (this time to the convex body $(1-\Delta)K$) we get that

$$(1-3\Delta)K \subseteq \text{int}_{\Delta r}((1-\Delta)K).$$

Thus, by Lemma B.2 we have that $\theta$ lies inside $\text{int}_{\Delta r}((1-\Delta)K)$ with probability at least $\frac{9}{10} - \delta \geq \frac{8}{10}$ (as $\theta$ is sampled from $\pi$ with TV error $\leq \delta$). Therefore, since $\xi \sim B(0, \Delta r)$, we must also have that the probability that the point $\hat{\theta} = \frac{1}{1-\Delta}(\theta + \xi)$ is in $K$ (and is therefore not rejected) is greater than $\frac{8}{10}$[4]. This implies that the number of iterations until our algorithm returns a point $\hat{\theta}$ is less than $k > 0$ with probability at least $1 - 2^{-k}$, and the expected number of iterations is at most 2 (proved in Corollary B.4).

Since each iteration requires one random sample $\theta$ from the distribution $\mu$, and one call to a membership oracle for $K$ (to determine if $\frac{1}{1-\Delta}Z \in K$), the number of sampling oracle and membership oracle calls required by Algorithm 1 is $O(1)$ with very high probability. Therefore, with high probability, Algorithm 1 returns a point $\hat{\theta}$ from a distribution with infinity-distance at most $\varepsilon$ from $\pi$ after $O(1)$ calls to the sampling and membership oracles.

Since Algorithm 1 succeeds with probability $1 - 2^{-k}$ after $k$ iterations, after

$$\tau_{\max} = 5d \log\left(\frac{R}{r}\right) + 5LR + \varepsilon$$

iterations Algorithm 1 will have succeeded with probability roughly $1 - \varepsilon(\frac{R}{r})^{-5d} e^{-5LR}$. In the very unlikely event that Algorithm 1 still has not succeeded after $\tau_{\max}$ iterations, Algorithm 1 simply outputs a point sampled from the uniform distribution on the ball $B(0, r)$ of radius $r$ contained in $K$. The probability mass of the target distribution $\pi$ inside this ball is at least as large as $(\frac{R}{r})^{-d} e^{-LR}$; thus, since $f$ is $L$-Lipschitz, we show in Corollary B.4 that outputing a sample from the uniform distribution on this ball with probability $\varepsilon(\frac{R}{r})^{-5d} e^{-5LR}$ does not change the $\infty$-distance error of the sample returned by the algorithm by more than $\varepsilon$.

---

[4]In Algorithm 1 we reject $\hat{\theta}$ with a slightly higher probability to ensure that, in differential privacy applications, in addition to the privacy of the point returned by the algorithm, the runtime is also $\varepsilon$-differentially private.

## 3.2 Completing the proof of Theorem 2.1

**Proof:** By Theorem 1, the output of Algorithm 1 has infinity-distance error bounded by $\varepsilon$ as long as the input samples have TV distance error bounded by

$$\delta \leq O\left(\varepsilon \times \left(\frac{R(d\log(R/r) + LR)^2}{\varepsilon r}\right)^{-d} e^{-LR}\right),$$

and, with high probability, Algorithm 1 requires $O(1)$ such independent samples. To generate a sample from $\pi$ with TV error $O(\delta)$ when $K = \{\theta \in \mathbb{R}^d : A\theta \leq b\}$ is a polytope defined by $m$ inequalities, we use the Dikin Walk Markov chain of [34]. This Markov chain requires an initial point from some distribution $\mu_0$ which is $w$-warm with respect to the stationary distribution $\pi$, that is, $\sup_{z \in K} \frac{\mu_0(z)}{\pi(z)} \leq w$. To obtain a warm start, we let $\mu_0$ be the uniform distribution on the ball with radius $r$ contained in $K$ and sample from $\mu_0$. Since $f$ is $L$-Lipschitz, and $K$ is contained in a ball of radius $R$, $\mu_0$ is $w$-warm with

$$w \leq \frac{1}{\text{Vol}(B(0,r))} \times \left(\frac{\max_{z \in K} \pi(\theta)}{\min_{z \in K} \pi(\theta)} \times \text{Vol}(B(0,R))\right) \leq \left(\frac{R}{r}\right)^d \times e^{RL}.$$

From [34], we have that from this $w$-warm start the Dikin Walk Markov chain requires at most $O((m^2 d^4 + m^2 d^2 L^2 R^2)\log(\frac{w}{\delta}))$ steps to generate a sample with TV distance at most $\delta$ from $\pi$, where each step makes one function evaluation and $O(md^{\omega-1})$ arithmetic operations. Plugging in the above values of $\delta, w$ the number of Markov chain steps is

$$T = O((m^2 d^3 + m^2 d L^2 R^2) \times [LR + d\log(\frac{Rd + LRd}{r\varepsilon})])$$

to generate each independent sample with the required TV error $O(\delta)$. Since the number of independent samples required as input for Algorithm 1 is $O(1)$ w.h.p., the number of arithmetic operations for Algorithm 1 to output a point with at most $\varepsilon$ infinity-distance error is $O(T \times md^{\omega-1})$.

Finally, we note that in the more general setting where $K$ is a convex body with membership oracle (but not necessarily) a polytope, we can instead use, for instance, the hit-and-run Markov chain of [29] to generate samples from $\pi$ with TV error $O(\delta)$ in a number of membership and function evaluation oracle calls that is polynomial in $d$ and poly-logarithmic in $\frac{1}{\delta}, R, r$. We can then plug this sample into our Algorithm 1 to obtain a sample from $\pi$ with infinity-distance error $O(\varepsilon)$ in a number of oracle calls that is (poly)-logarithmic in $\frac{1}{\varepsilon}, \frac{1}{r}$ and polynomial on $d, L, R$. (see Remark 2.3). ∎

## 4 Conclusion, Limitations, and Future Work

To the best of our knowledge, this is the first work that presents an algorithm for sampling from log-concave distributions on convex bodies that comes with infinity-distance bounds and whose running time depends logarithmically on $1/\varepsilon$. Towards this, the main technical contribution is Algorithm 1 (and Theorem 2.2) which achieves this improved dependence on $\varepsilon$ by taking as input continuous samples from a convex body with TV bounds and converting them to samples with infinity-distance bounds.

On the other hand, our bounds are polynomial in $LR$, yet there are algorithms for sampling from logconcave distributions $\pi \propto e^{-f}$ on a convex body in the total variation distance that are poly-logarithmic in $R$ and do not assume $f$ to be Lipschitz [29]. Thus, the main open problem that remains is whether one can also obtain running time bounds for sampling in the infinity-distance which are poly-logarithmic in $R$ and do not require $f$ to be Lipschitz.

Our main result also has direct applications to differentially private optimization (Corollaries 2.4 and 2.5). Differential privacy is a notion which has been embraced in many technologies in societal contexts where privacy of individuals is a concern. Hence, we see our work to have a potential of positive societal impact and do not foresee any potential negative societal impacts.

## Acknowledgments and Disclosure of Funding

This research was supported in part by NSF CCF-2104528, CCF-1908347, and CCF-2112665 awards.

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
