# OpenReview forum: "Sampling from Log-Concave Distributions with Infinity-Distance Guarantees"
_NeurIPS.cc/2022/Conference — NeurIPS 2022 Accept_

### Official Review · Reviewer_HEwz · 2022-07-05

**Rating:** 7
**Confidence:** 3
**Soundness:** 4 excellent
**Presentation:** 3 good
**Contribution:** 3 good

**Summary:**

The main aim of the paper is to introduce a sampling scheme with guarantees in the infinity distance when the sampled measure is log-concave.  Several applications and consequences, in differential privacy, are then demonstrated.

**Questions:**

Question/concern about Corollary 2.5:
In this application, there is a convex function $f(\theta) = \sum \ell_i(\theta, x_i)$ where $x_i \in \mathbb{R}^n$ and the task is to sample from $\pi \propto e^{-f(\theta)/ \epsilon}d\theta$. The solution is to use the Dikin random walk to sample from $\pi$ in total variation and then to use the above procedure to upgrade the guarantee to the infinity distance. The claim is that the output is differentially private.

Now, I certainly agree that any point sampled from $\pi$ (or something close to it) is differentially private. However, the Dikin random walk proceeds by evaluating the value of $f(\theta)$ at each iteration, and the value of $f$ *depends* on the sample $\{x_i\}$.
So each iteration of the Dikin random walk is not private, and reveals something about the function being optimized and hence the sample.

Moreover, I'm not sure I agree with the statement "Otherwise, we output
$\hat{\theta}= 0 \in K$, which is clearly $\epsilon$-differentially private since this output does not depend on the data". If the algorithm fails to stop it is more likely that the sampled points were near the boundary of the body. The probability of this happening depends on the function being optimized, which again depends on the sample.

As I indicated above, I am not an expert in differential privacy. If I missed something basic I would appreciate the authors' explanation.

Minor comments:
- Line 58: "bounds in the Wasserstein distance, KL divergence, and Renyi divergence metrics also do
59 not imply bounds on the infinity distance," Isn't the Infinity distance simply the infinity Renyi divergence?

- Theorem 2.2: $f$ should be declared to be convex.

- Even after reading Remark 2.3 I find it very weird that Theorem 2.1 is stated with respect to polytopes and Theorem 2.2 is with respect to arbitrary convex bodies.

- Corollary 2.2: Again, $f$ should be declared to convex.

- Figure 1 really belongs in the appendix where it gives a visual aid to the proof.

- Corollary B.4: This is not really important, but perhaps it is worth mentioning that the stopping time $\tau$ has a (or more precisely dominated by) a geometric distribution. This immediately gives the tail and expectation bounds.

- Line 728: Is $y$ distributed like $\mu$ or $\hat{\mu}$ (it is $\mu$ in the paragraph preceding the statement of the Lemma). It might be a good idea to remind the readers of all $\mu$ and $\nu$ notation before stating the lemma.

- The authors make repeated use of the trivial inequality $\max \pi(\theta) *Vol(B(0,R)) \geq 1$. It could be helpful to state it somewhere at the beginning of the section and refer back to it,

- Line 731-732: I think there is an offset in the 'numbering' of the inequalities.

- Line 760: Should $B(0,r)$ be $B(a,r)$?

---------------------------------------------------------------------------------------------------------------------------
Post Rebuttal: The authors addressed my concerns and I raised my score.

**Limitations:**

 The authors adequately addressed the limitations and potential negative societal impact of their work

**Strengths And Weaknesses:**

The problem of sampling in the so-called 'infinity distance' is an interesting problem that has gained some attention recently, mostly because of its potential applications to differential privacy. In this respect, the paper makes an interesting contribution that should be of interest to the NeurIPS community.

One major comment from my side is that this is not a paper about sampling. The main result is a somewhat simple and intuitive observation that shows how to 'upgrade' guarantees in total variation to guarantees in the infinity distance.
With slightly more details, suppose that one wishes to sample from the log-concave measure $\pi$ and that one was able to attain a sample from $X \sim \mu$ which is close to $\pi$ in total variation. Then, by smoothing $\mu$ (with a convolution kernel), stretching it (by multiplying $X$ by a constant), and then conditioning it back to the support of $\pi$ one gets a new law whose density is similar to $\pi$ almost everywhere. From the description, it is clear that this procedure can be made algorithmic and it is shown that as long as $\pi$ also has a log-Lipschitz density then the infinity distance guarantees are good.

Given my criticism above, I wish to emphasize one point. Even if the observation seems to be simple, it was not observed before, and its simplicity could be seen as an advantage, both since it is easily implementable and because it is easy to understand. I read through most of the proofs and could follow everything easily (I did find the informal explanation in the main body to be a bit confusing).

I have one concern, which I detail below, related to one of the applications. It could be that the concern comes from my lack of expertise in differential privacy, and I am prepared to raise my score after reading the author's response.

---

> ### Author Response · Authors · 2022-08-02
> **Response to Reviewer HEwz**
>
> Thank you for your valuable comments and suggestions. We are encouraged that you find our paper to be an interesting contribution to the NeurIPS community. We answer your specific questions below and we thank you for offering to increase your score if your concerns are addressed.
>
> *``I certainly agree that any point sampled from $\pi$ (or something close to it) is differentially private.
> However, the Dikin random walk proceeds by evaluating the value of $f(\theta)$ at each iteration...''*
>
> While it is indeed true that each step of the Dikin random walk depends on the data, this does not imply that the output of our algorithm is not private. This is because the intermediate iterations of the Dikin walk are computed by a trusted server which has direct access to the dataset, and these iterations are not released to the public. Only the last iteration of the Dikin walk, which is close to the target distribution $\pi$ in TV distance, is used by our Algorithm 1 to compute the output point. This output point is $\varepsilon$-close to the target distribution $\pi$ in the infinity-distance metric, and therefore our algorithm, which outputs only this single point to the public, is $\varepsilon$-differentially private. The assumption that the Markov chain is computed by a trusted server, and only the last step of the Markov chain is used to generate the output point which is released to the public, is a standard assumption in the differential privacy literature (see e.g., [Bassily,  Smith, Thakurta, FOCS 2014]).
>
>
> *``I'm not sure I agree with the statement "Otherwise, we output $\hat{\theta}\in K$, which is clearly $\varepsilon$-differentially private since this output does not depend on the data". If the algorithm fails to stop it is more likely that the sampled points were near the boundary of the body...''*
>
> Sorry for the confusion--perhaps this sentence needs more explanation. It is true that, while the point $0 \in K$ does not depend on the data,  knowledge of the *probability* $p := P(\mathrm{Output} =0|x)$ that our Algorithm 1 rejects all $\tau_{\mathrm{max}}$ samples and outputs the point $0 \in K$ does depend on the data $x$. To deal with this issue and to ensure that knowledge of this probability $p$ does not compromise privacy, our Algorithm 1 (in Step 7) rejects the proposed point $\hat{\theta}$ with probability at least $\frac{1}{2}$ even if the proposed point  $\hat{\theta}$ is contained in $K$ (the rejection probability is $\frac{1}{2}$ if $\hat{\theta} \in K$ and $1$ if $\hat{\theta} \notin K$). Roughly speaking, this ensures that the probability $p$ that our algorithm rejects at all $\tau_{\mathrm{max}}$ iterations and outputs $0 \in K$ does not depend on the data by more than a factor of $e^{\varepsilon}$. More specifically, we show in the proof of Theorem 2.2 (the equations following Equation (4) in the original paper's appendix, labeled as Equation (5) in the revised version) that the probability that Algorithm 1 will reject at all $\tau_{\mathrm{max}}$ iterations satisfies $(1/2)^{\tau_{\mathrm{max}}}e^{-\varepsilon/2} \leq p \leq (1/2)^{\tau_{\mathrm{max}}}e^{\varepsilon/2}$, regardless of which dataset $x$ we use. Thus, the probability $p = P(\mathrm{Output} =0 |x)$ that our algorithm rejects at all $\tau_{\mathrm{max}}$ iterations and outputs the point $0 \in K$ satisfies $P(\mathrm{Output} =0| x ) \leq e^{\varepsilon} P(\mathrm{Output} = 0| x')$ for any two datasets $x,x'$, which ensures that our algorithm satisfies the definition of $\varepsilon$-differential privacy even when it outputs the point $0 \in K$. In the revised version of our paper (Lines 815-823 in the proof of Corollary 2.4), we provide a detailed explanation of why our algorithm satisfies $\varepsilon$-differential privacy even when it outputs the point $0 \in K$ (see also Remark C.2 in either version of the paper).
>
> *Minor comments:*  Thank you for pointing us to these typos. We addressed them in the revised paper.
>
> For the minor comment ``Theorem 2.2: $f$ should be declared to be convex,'' we note that the intermediate result in Theorem 2.2 only requires $f$ to be Lipschitz. However, to apply Theorem 2.2 when proving Theorem 2.1 we need to provide Algorithm 1 with a point which is close to $\pi$ in TV distance, and convexity allows us to generate this input point efficiently via a Markov chain algorithm.
>
> For the minor comment ``Theorem 2.1 is stated with respect to polytopes and Theorem 2.2 is with respect to arbitrary convex bodies,'' we note that, to prove Theorem 2.1, we provide Theorem 2.2 with a point generated by the Dikin walk Markov chain which is close to $\pi$ in the TV distance. To implement the Dikin walk Markov chain we need the convex body to be a polytope. While we could have instead used a different Markov chain, like the hit-and-run Markov chain, which only requires $K$ to be a convex body, the runtime bounds we would have obtained would not be as fast as those we obtain in Theorem 2.1 via the Dikin walk.

---

> > ### Comment · Reviewer_HEwz · 2022-08-03
> > **Response to rebuttal**
> >
> > Thank you for the clarifications, which addressed all of my concerns. I apologize for missing Remark C.2.
> >
> > As I said, this is a solid contribution, and I am happy to raise my score.

---

### Official Review · Reviewer_JWN3 · 2022-07-09

**Rating:** 7
**Confidence:** 3
**Soundness:** 4 excellent
**Presentation:** 3 good
**Contribution:** 3 good

**Summary:**

The authors propose an approximate sampling algorithm for compactly supported, log-concave, Lipschitz distributions. They prove that the distribution of the sample output by their algorithm is guaranteed to be $\epsilon$-close in *infinity distance* to the target distribution in $\mathcal{O}(\mathrm{poly}(\log\epsilon^{-1}, d))$ steps, where $d_\infty(q, p) = \sup_x \lvert \log q(x) - \log p(x) \rvert$.

The key ingredient of their algorithm is a rejection sampler that takes an approximate sample whose law is already $\delta$-close to the target in TV distance for some appropriately chosen $\delta$. Then, it outputs another approximate sample with the required $\epsilon$ infinity distance guarantee. They note that they can use previous methods that require $\mathcal{O}(\mathrm{poly}(\log\epsilon^{-1}, d))$ steps to attain a sample whose law is within the required $\delta$ TV bound. Hence, combining it with their rejection sampler gives the desired result.

The authors also show that their method immediately yields more efficient solutions to some previously studied differentially privacy problems.

**Questions:**



**Limitations:**



**Strengths And Weaknesses:**

# Strengths
The authors study a relevant problem and propose an elegant solution to it that significantly improves the best known asymptotic runtime for the problem, from $\mathcal{O}(\epsilon^{-2}\mathrm{poly}(d))$ to $\mathcal{O}(\mathrm{poly}(\log\epsilon^{-1}, d))$ steps. I appreciated that the authors also give some intuition in the main text and a lot more in Appendix A regarding why extending previous methods won't yield a solution as good as theirs.

The rejection sampler is a nice, simple algorithm and the proofs showing its desired properties are nicely laid out and clearly written. I checked the proofs in Appendix B and believe they are all correct.

Altogether, I think the paper is a solid contribution to the literature.

# Weaknesses

The greatest weakness of the paper is the lack of experiments, even in the supplementary material. I think it would be particularly nice if the authors could construct a distribution that is within the required TV distance to an appropriate target but far in terms of infinity distance and run their rejection sampler. Then, they could show that within the expected number of steps, the histogram of the samples output by their algorithm is within the required infinity distance to the target.

The second weakness is that I found the description of the second DP problem unnecessarily technical and hard to follow, in contrast with the rest of the paper. I would either make its description higher level or give more detail, as currently, it is quite hard to follow without a background in DP.

## Miscellaneous & Typos
 - The whole paper, including the main text, has been uploaded as the supplementary material.
 - There is a slight inconsistency in the exposition in Section 2: the authors allow the inner ball to be centred at an arbitrary point $a$, while in Alg 1 and Section 3, they require the inner ball to be centred at the origin. This is especially confusing, as they later reuse $a$ as the distance between certain grid cells on L220 in Section 3.1. However, I believe that this shouldn't change any of their results.
 - I believe the lower bound on the infinity distance on L257 should be $d\log 2$ instead of $2^d$.
 - What is the $E_P[\cdot]$ operator on L179 and L189?

---

> ### Author Response · Authors · 2022-08-02
> **Response to Reviewer JWN3**
>
> Thank you for your valuable comments and suggestions. We are glad you find our paper to be a solid contribution to the literature. We answer your specific questions below.
>
> *``it would be particularly nice if the authors could construct a distribution...''*
>
> Thank you for the suggestion. In Appendix D of our revised paper, we implement Algorithm 1 on two distributions: a simple one-dimensional distribution, and a $d=100$ dimensional ``Dirichlet" distribution.
>
> *One-dimensional distribution:* We first apply our algorithm to the one-dimensional distribution  $\pi(\theta) = e^{\frac{1}{2}(3-\theta)}$ with support on $K = [-1,3]$, and compute the histogram and infinity-distance error of the distribution $\nu$ of its output (see https://github.com/hcjc42zxai83/P1/files/9238437/histogram.pdf, or Figure 2 in Appendix D of revised paper). We observe that, when provided with samples from a distribution $\mu$ which is close to $\pi$ in TV distance (TV distance $\leq \frac{1}{100}$), but far from $\pi$ in infinity distance ($d_{\infty}(\pi,\mu) = \infty$), our algorithm (with parameters $\varepsilon = 0.1$, $L= \frac{1}{2}$, $R=4$, and $\Delta = \frac{\varepsilon}{\max(d,LR)}= 0.05$) returns points from a distribution $\nu$ which (as computed with the histogram) is close to $\pi$ in infinity distance with $\mathrm{d}_{\infty}(\nu,\pi)=0.1054$, which roughly matches the parameter $\varepsilon=0.1$ we set in our algorithm. Moreover, we observe the average number $\tau$ of iterations required by our algorithm is only 2.1904 (averaged over $10^7$ runs), satisfying the guarantee $E[\tau]\leq 3$ given by our Theorem 2.2 (see Footnote 3 in either version for this guarantee).
>
> *$100$-dimensional Dirichlet distribution:* We also implement Algorithm 1 on a $d=100$ dimensional ``Dirichlet" distribution $\pi(\theta) \propto \prod_{i=1}^d \theta_i$ with support on the simplex $K =$ { $\theta \in R^d:  \sum_{i=1}^d \theta_i \leq 1, \theta_i \in [0,1] \forall i \in [d]$}. We observe that, when we provide our Algorithm 1 with samples from a distribution $\mu$ which is close to $\pi$ in TV distance (TV distance $\leq 10^{-d}$), but far from $\pi$ in infinity distance ($\mathrm{d}_{\infty}(\pi,\mu)=\infty$), Algorithm 1 terminates after an average of 1.9935 iterations (averaged  over $10^5$ runs), thus satisfying the guarantee $E[\tau]\leq 3$ of our Theorem 2.2. (We do not compute the histogram and infinity distance for the $d=100$ dimensional Dirichlet distribution, since the number of points needed to compute the histogram grows exponentially with $d$.)
>
> *``Description of the second DP problem...''*
>
>  We are sorry you found our description confusing, and have added additional background and explanations on Page 5 of our revised paper. Namely, given a sample covariance matrix $\Sigma$ and  $k>0$, the goal is to find a (random) rank-$k$ projection matrix $P$ which maximizes the average variance $E_P[\langle\Sigma,P\rangle]$ of $\Sigma$ (also referred to as the utility of $P$), under the constraint that $P$ is $\varepsilon$-differentially private (DP) ($E_P$ denotes expectation w.r.t. the random matrix $P$). This problem has many ML applications, including DP principal component analysis, and was previously studied in e.g. [Chaudhuri, Sarwate, Sinha, NeurIPS 2012] and [Hardt, Price, NeurIPS 2014].
>
> When privacy is not a concern, the $P$ which maximizes the variance is a projection matrix onto the subspace spanned by top-$k$ eigenvectors of $\Sigma$, with maximum variance $\langle \Sigma, P \rangle=\sum_{i=1}^k\lambda_i$, where $\lambda_1\geq\cdots\geq \lambda_d$ denote the eigenvalues of $\Sigma$. However, when privacy is a concern, there is a tradeoff between the privacy level $\varepsilon$ and utility $E_P[\langle\Sigma,P\rangle]$-- the maximum utility one can achieve decreases with $\varepsilon$.
>
> The best current utility bound for $\varepsilon$-DP low rank approximation was achieved in [Leake, McSwiggen, Vishnoi, STOC 2020], who show one can find a pure $\varepsilon$-DP  rank-$k$ projection matrix $P$ such that $E_P[\langle \Sigma, P \rangle] \geq (1-\delta)\sum_{i=1}^k \lambda_i$ whenever  $\sum_{i=1}^k\lambda_i\geq\Omega(\frac{dk}{\epsilon\delta}\log\frac{1}{\delta})$ for any $\delta>0$. To generate $P$, their algorithm generates a sample, with infinity-distance error $O(\varepsilon)$, from a Lipschitz concave log-density on a polytope, and transforms this sample into a projection matrix $P$. The runtime of their sampling algorithm is polynomial in $\frac{1}{\varepsilon}$ and $d$. In our Corollary 2.5 we show that, by replacing their sampling algorithm with the algorithm in our Theorem 2.1, we obtain an $\varepsilon$-DP algorithm for low rank approximation which achieves the same optimal utility bound, yet has improved runtime logarithmic in $\frac{1}{\varepsilon}$ and polynomial in $d$.
>
> *``Miscellaneous & Typos''*
>
>  Thank you for pointing out these typos. We corrected them in the revised version.

---

> > ### Comment · Reviewer_JWN3 · 2022-08-04
> > **Response to the authors**
> >
> > I thank the authors for their elaborate response. I maintain that their work is a strong contribution to the literature and I keep my score.

---

### Official Review · Reviewer_YRWJ · 2022-07-11

**Rating:** 8
**Confidence:** 3
**Soundness:** 4 excellent
**Presentation:** 4 excellent
**Contribution:** 4 excellent

**Summary:**

This work gives the first algorithm (or rather "meta-algorithm") for sampling from log-concave distributions with an $\epsilon$ infinity-distance guarantee and runtime which only has a poly-logarithmic dependence on $1/\epsilon$. In particular, this paper provides a technique to convert any sampling algorithm with TV distance guarantees to one with infinity-distance guarantees. Further, they provide applications of their result in differentially private optimization i.e. differentially private empirical risk minimization and differentially private low-rank approximation.

**Questions:**

Since the paper has strong theoretical contributions, I am okay with it not having any experiments. But it would be interesting to see how the current algorithm (it looks simple to implement) would compare in practice to previous algorithms with infinity-distance guarantees on well-known log-concave distributions.

**Limitations:**

Yes.

**Strengths And Weaknesses:**

Originality: I am not an expert in this area so I am not entirely sure about other related work. From reading the paper, this is a clear advance over previous state-of-the-art in sampling from log-concave distributions with infinity-distance guarantees.

Quality: The submission is technically sound. All claims are well-supported with proofs.

Clarity: The submission is clearly written and well-organized.

Significance: Yes, the problems being studied are quite significant. Log-concave distributions are extremely well-studied in the literature with lots of applications. In particular, sampling from log-concave distributions with infinity-distance guarantees has applications in differential privacy, as shown in the applications in this work. Even otherwise, it is a very natural question to study. Improvement from runtimes that are poly(1/$\epsilon$) to poly-log(1/$\epsilon$) is an exponential improvement and so it is quite significant.

---

> ### Author Response · Authors · 2022-08-02
> **Response to Reviewer YRWJ**
>
> Thank you for your valuable comments and suggestions.  We are glad you find our work to be a clear advance over previous state-of-the-art in sampling from log-concave distributions with infinity-distance guarantees. We answer your specific question below.
>
> *``Since the paper has strong theoretical contributions, I am okay with it not having any experiments. But it would be interesting to see how the current algorithm (it looks simple to implement) would compare in practice to previous algorithms…''*
>
> Thank you for the suggestion— our algorithm is indeed simple to implement, and it would be interesting to see how our algorithm performs in experiments.
>
> As a first step, in Appendix D of the revised version of our paper, we have implemented our post-processing algorithm (Algorithm 1), on simple test distributions which can be approximately sampled without running a  Markov chain. We would be happy to include experiments which combine our Algorithm with the Dikin walk Markov chain, as well as experiments comparing the performance of our algorithm to the grid walk Markov chain algorithm which was previously used in [Bassily, Smith, Thakurta, FOCS 2014] to sample from log-concave distributions with infinity distance guarantees, in the final version of our paper.

---

### Official Review · Reviewer_JJsb · 2022-07-12

**Rating:** 7
**Confidence:** 3
**Soundness:** 3 good
**Presentation:** 3 good
**Contribution:** 2 fair

**Summary:**

This paper proposes an algorithm to approximately sample from a log-concave distribution over a convex continuous body. In applications like Bayesian statistics, sampling exactly from a distribution often turns out to be difficult because its explicit form is known only up to proportionality. Also, one legacy application is to integrate a log-concave log-Lipschitz function over a convex body. One common practice to tackle such problems is to construct a Markov Chain over the discretized constraint space. Although such a method can generate samples from a distribution $\epsilon$ close in total variation distance to the true distribution in computation time $polylog(1/\epsilon)$, it fails to give a similar computation time algorithm to generate samples from  $\epsilon$ close distribution in infinity-distance. The proposed sampling algorithm in this paper is able to give $poly(\log 1/\epsilon, d)$ guarantee on runtime to generate $\epsilon$ infinity distance samples. The result also improves upon the dependency on dimension d in comparison to the state of the art. The result has its application in optimization preserving differential privacy in mechanism design game theory literature. The novelty of the method proposed here is to take a sample which is $O(\epsilon)$ close in TV distance and convert it to $O(\epsilon)$ close sample in infinity distance. It effectively uses a continuous-space Markov chain to bypass the limitation of the previous methods. The simulation of the continuous-space Markov chain has been done by convolving the initial sample with uniform noise followed by rejection sampling.

The paper is well written and well organized.

Relevance: The result is quite relevant the authors have shown its applications in differentially private optimization. Nevertheless, approximate sampling from log-concave distributions with various distance guarantees has been at the core of learning literature

Correctness: The authors have given proof ideas of their main theorems. Although I have not been through the minute details, overall they seem to be correct.


**Questions:**

None.

**Limitations:**

None.

**Strengths And Weaknesses:**

The work is original, clear, and relevant.

---

> ### Author Response · Authors · 2022-08-02
> **Response to Reviewer JJsb**
>
> Thank you for your valuable comments, and for taking the time to review our paper.  We are glad you find our work to be original, clear, and relevant, and we thank you for supporting our paper.

---

### Meta-Review · Area_Chair_iYWT · 2022-08-25

**Recommendation:** Accept
**Confidence:** Certain

**Metareview:**

Sampling from log-concave distributions is a well studied problem and there are many existing algorithms that can sample from a distribution close to the true distribution up to a small total variation distance. The paper gives a new reduction that can use these algorithms as a subroutine to get samples from a distribution close to the true distribution in infinity distance i.e. the densities are close everywhere. This problem commonly arises in differentially private optimization. The reduction is simple and can be implemented easily. All reviewers agree that the paper is a significant contribution to the literature, it is well written, and the algorithm has potential to be useful in practice.

**Award:**

No

---

### Decision · Program_Chairs · 2022-09-14

Accept